# Spatial distribution and determinants of HIV prevalence among adults in urban Ethiopia: Findings from the Ethiopia Population-based HIV Impact Assessment Survey (2017–2018)

Terefe Gelibo[1]*, Sileshi Lulseged[1], Frehywot Eshetu[2], Saro Abdella[3‡],
Zenebe Melaku[1], Solape Ajiboye[4‡], Minilik Demissie[3‡], Chelsea Solmo[5‡],
Jelaludin Ahmed[2‡], Yimam Getaneh[3], Susan C. Kaydos-Daniels[2‡], Ebba Abate[3‡],
EPHIA Study Group[¶]

1 ICAP in Ethiopia, Mailman School of Public Health, Columbia University, Addis Ababa, Ethiopia, 2 Division of Global HIV & TB (DGHT), United States Centers for Disease Control and Prevention (CDC), Addis Ababa, Ethiopia, 3 TB/HIV Directorate, Ethiopia Public Health Institute (EPHI), Addis Ababa, Ethiopia, 4 Division of Global HIV & TB (DGHT), United States Centers for Disease Control and Prevention (CDC), Atlanta, GA, United States of America, 5 ICAP at Columbia University, New York, New York, United States of America

☯ These authors contributed equally to this work.
‡ SA, SA, MD, CS, JD, SCKD and EA also contributed equally to this work.
¶ Membership of the EPHIA Study Group is provided in the S1 File.
* mamater.1986@gmail.com

**Data Availability Statement:** All the EPHIA data files are available from the PHIA Project database at

## Abstract

The design and evaluation of national HIV programs often rely on aggregated national data, which may obscure localized HIV epidemics. In Ethiopia, even though the national adult HIV prevalence has decreased, little information is available about local areas and subpopulations. To inform HIV prevention efforts for specific populations, we identified geographic locations and drivers of HIV transmission. We used data from adults aged 15–64 years who participated in the Ethiopian Population-based HIV Impact Assessment survey (October 2017–April 2018). Location-related information for the survey clusters was obtained from the 2007 Ethiopia population census. Spatial autocorrelation of HIV prevalence data were analyzed via a Global Moran's I test. Geographically weighted regression analysis was used to show the relationship of covariates. The finding indicated that uncircumcised men in certain hotspot towns and divorced or widowed individuals in hotspot woredas/towns might have contributed to the average increase in HIV prevalence in the hotspot areas. Hotspot analysis findings indicated that, localized, context-specific intervention efforts tailored to at-risk populations, such as divorced or widowed women or uncircumcised men, could decrease HIV transmission and prevalence in urban Ethiopia.

## Introduction

In resource-limited settings, such as sub-Saharan Africa, sustained HIV interventions targeting all populations are not feasible [1, 2]. Understanding the HIV epidemic at the local level and

https://phia-data.icap.columbia.edu/datasets?country_id=12.

**Funding:** The author(s) received no specific funding for this work.

**Competing interests:** The authors have declared that no competing interests exist.

reallocating resources for its control in specific areas in countries with a generalized epidemic has been recommended to increase cost-effectiveness [2, 3]. Identifying HIV clusters can inform tailored HIV interventions [3].

Ethiopian Population-based HIV Impact Assessment (EPHIA) survey data suggest that the incidence of HIV in urban Ethiopia is around 0.5% [4], much lower than the Joint United Nations Programme on HIV/AIDS benchmark (3%) [5]. Even though adult HIV prevalence, or the proportion of persons in a population who are living with HIV at a specific point in time, has declined at the national level, little information is available about sub-geographic areas and certain subpopulations in urban Ethiopia. Analysis of data from the 2011–2016 demographic and health survey in Ethiopia indicated that the distribution of HIV infection in Ethiopia is not random [6–8]. A study in one region in Ethiopia revealed that low educational status and migration status are determinants of HIV infection in Ethiopia [9]. Moreover, HIV service coverage varies among subpopulations and locations [10]. The HIV epidemic in certain localities could emerge or re-emerge if not addressed in these sub-regions and subpopulations [6].

Spatial analysis in epidemiology links spatial data (either from an absolute location and/or relative spatial arrangement of the data) to disease spread or high-risk populations [11] and has been applied to HIV intervention research in Africa [12]. Geospatial analysis of epidemiological data can generate precise maps of hotspot locations where HIV prevalence is concentrated [13]. Local spatial analyses can show geographic variation of the HIV epidemic and its drivers and inform targeted interventions; however, few geospatial analyses use data from subpopulations in Ethiopia [6–9].

With a heterogeneous HIV burden and decreasing national HIV prevalence, Ethiopia's epidemic is primarily concentrated in certain populations [7]. Low national HIV prevalence may obscure localized epidemics in urban Ethiopia, and identifying hotspot areas could help target these high-risk populations. The purpose of this study is to identify geospatial clustering of HIV infections and hotspot areas by subpopulation groups to inform targeted interventions with the limited resources available.

## Methods

### Study design and population

Our geospatial analyses used data from adults who participated in the EPHIA survey (October 2017–April 2018) [14]. EPHIA was a nationally and regionally representative, cross-sectional, household-based sero-survey conducted among 19,136 adults aged 15–64 years and 4,729 children aged 0–14 years. The reference population comprised individuals in urban areas who were present in households (i.e., "slept in the household") on the night before the interview. All individuals in either the de facto (i.e., those who slept in the household the night before the survey) or de jure (i.e., those individuals who are usual residents of the household regardless of whether they were present in the household during the previous night) populations were included in the rosters compiled for sampling purposes, although our analysis was limited to the de facto population only. According to the population projection made from 2007 to 2037, currently 20.4% of Ethiopia's population lives in urban areas [15].

### Study variables and measurement

The primary outcome variable for the analysis was geo-linked HIV prevalence. HIV prevalence in each cluster was calculated from the HIV test results obtained in the survey (the denominator is the total population tested for HIV, and the numerator is the number of HIV-positive persons identified in this study). HIV testing was conducted in each household using the

Ethiopian National HIV testing algorithm in accordance with national guidelines. Individuals with a nonreactive result on the screening test were reported as HIV negative. Individuals with a reactive screening test underwent confirmatory testing. Those with reactive results on both the screening and confirmatory tests were classified as HIV positive. [4]. The HIV prevalence in each town was the outcome variable. HIV prevalence was defined as HIV infection on the day the blood sample was taken and HIV testing was conducted. The independent variables included survey-weighted proportions of women, mean age of all participants, being divorced or widowed, being uncircumcised, being unaware of HIV-positive status during EPHIA, having no regular sexual partner, not using condoms, having had sex before age 15 years, and using injected drugs. These socioeconomic, demographic, and biological variables were selected for this analysis because these factors have been associated with the spatial distribution of HIV and risk of HIV infection in other studies [16–24]. Data were aggregated at the town level.

## Sample size and sampling procedures

The EPHIA methodology, including data collection procedures, has been previously reported [14]. Briefly, the survey used a two-stage, stratified sampling design to provide national estimates from 11 regions across the country. In the first stage, 395 enumeration areas (EAs) were selected, and 393 EAs were included in the EPHIA survey (Fig 1).

The first-stage or primary sampling units for EPHIA are defined as EAs created for the 2007 Ethiopia Population and Housing Census. The 2007 sampling frame consisted of slightly over 17,000 urban EAs containing an estimated 3.0 million households [15]. EAs in six of the nine zones in the Somali region (Wogob, Jarar, Shebelle, Afder, Korahe, and Doolo) were excluded from the sampling frame for security reasons in the EPHIA survey. From each EA, a random sample of 30 households, on average, was selected for a total of 11,810 households. At each stage of the process, consent was indicated by signing or making a mark on the consent form on a tablet and on a printed copy, which was retained by the participant. After a designated head of household provided written consent for household members to participate in the survey, individual members were rostered during a household interview. Of the 20,170 adults who participated in the survey, 19,136 provided consent to be tested for HIV. Participants aged 15–64 years and emancipated minors ages 13–17 years provided written consent on a tablet for an interview and for participation in the biomarker component of the survey, including home-based testing and counselling, with return of HIV-test results as well as participation future research.

## Data collection

EPHIA staff used a questionnaire prepared for the survey to collect data about household and individual characteristics. Initial household-based HIV testing was performed with the national HIV-testing algorithm using three HIV rapid tests. Individuals with a reactive screening test underwent confirmatory testing using the Uni-Gold HIV-1/2 (Trinity Biotech, Bray, Ireland). Individuals with discordant results were administered a tiebreaker test (Vikia HIV-1/2, bioMérieux, Marcy-l'Étoile, France). All HIV-positive individuals identified in the field received confirmatory testing in a satellite laboratory using the Geenius HIV 1/2 Supplemental Assay (Bio-Rad, Hercules, CA USA) [4]. Both the questionnaire and field laboratory data were collected on mobile tablet devices using the application Open Data Kit, an open-source mobile data collection application. The Global Positioning System was used to identify and record the geographical coordinates of each EPHIA sample location. Cluster geolocation data were loaded onto encrypted and passcode-protected tablet computers in keyhole markup language

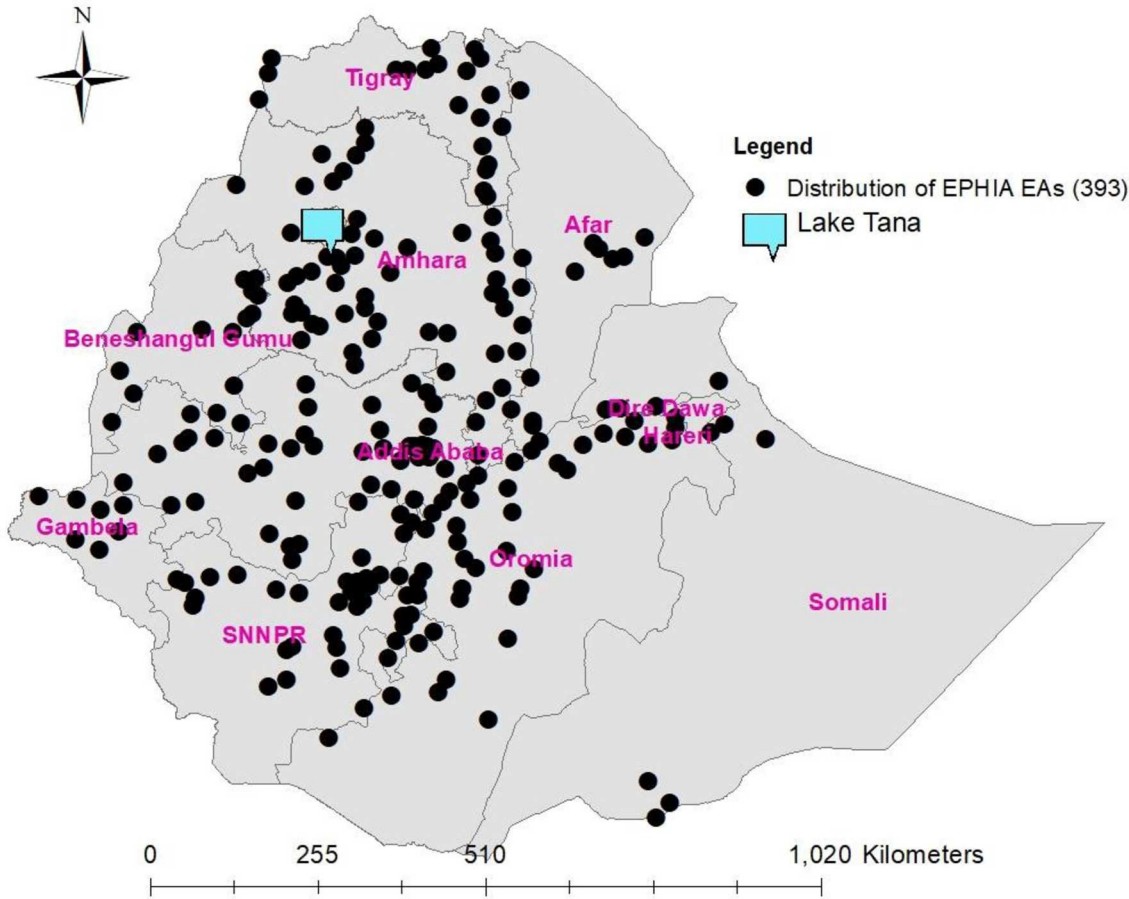

**Fig 1. Distribution of enumeration areas (N = 393) selected for Ethiopia Population-based HIV Impact Assessment Survey (2017–2018).**

format using the MAPS.ME mobile app (https://maps.me), which includes OpenStreetMap (**https://www.openstreetmap.org**) **data** [**25**].

## Data extraction

The 2017 EPHIA geospatial data sets were downloaded in Stata format with permission from the PHIA project website (at https://phia-data.icap.columbia.edu/datasets?country_id=12) [26]. After understanding the detailed data sets and coding, further data recoding was carried out. Data sets of Global Positioning System (GPS) coordinates of EPHIA clusters were merged for this analysis as appropriate per the Reference Guide for Using Geospatial Data from the Population-based HIV Impact Assessments [25].

## Analysis

All households and individuals within a dataset were assigned the location of the centroid of their respective survey EA, and then the centroid was randomly displaced within an area around the actual center of the EA [25].

Interpolation maps of HIV prevalence were created through ordinary kriging.. Kriging is a statistical model that infers the dependence between pairs of points by examining all similarly distant pairs in the data [27]. A hotspot analysis was conducted using Getis-Ord statistic.

ArcGIS (v.10.2, Esri, Redlands, CA USA) was used to detect statistically significant clusters of high or low HIV prevalence.

The spatial autocorrelation of HIV prevalence data was analyzed by performing a global Moran's I statistic to test the null hypothesis that observed data at one location are independent of data at other locations. Moran I is a measure of spatial autocorrelation in area data, explaining the degree of dependence between values of a variable at different geographic locations [28].

Because the Moran I statistic showed significant spatial autocorrelation of our HIV prevalence data (Moran I, 0.057; $p = 0.00014$) within a distance of 150 km., we subsequently analyzed the data using kriging [27]. The enumeration area (EA) in Ethiopia were used as the spatial mapping unit at the woreda/town level in this study which comprises about 249 woreda in this analysis, since the woreda is the basic decentralized administrative unit [29], and the woreda health system is the primary level of health care provided in Ethiopia [30]. These basic units were used based on previous studies [31], which suggested that spatial analysis results should be aggregated to the lower levels of services or programs relevant for decision-making, budgeting, and monitoring (including the district or facility levels).

To explore the correlations between covariates and HIV prevalence, we used Pearson correlation coefficient and geographically weighted regression (GWR) analysis. GWR) is a widely used tool for exploring spatial heterogeneity of conditions over geographic space [32]. Covariates that had low correlation (with correlation coefficient [r] <0.2) with HIV prevalence were not included in the models. Highly correlated covariates were not simultaneously included in the regression models to avoid model redundancy and multicollinearity. We used an explanatory approach and used the maps of hotspots of HIV prevalence to select covariates of interest ($p<0.05$). The prevalence of HIV in the hotspot towns was significantly correlated with the percentage of women who were not virally suppressed; individuals who were widowed, divorced, or separated; uncircumcised men; and individuals who lived in women-headed households during the survey period.

Following correlation analysis, we fitted an ordinary least squares (OLS) model to explain the global relationship between HIV prevalence and the covariates using a sample of 249 towns (observations). The outcome variable for the regression model (OLS) was the weighted HIV prevalence in each town.

$$HIVi = \beta 0 + \beta 1i + \beta 2i + \beta 3i + \beta 4i + \beta 5i + \beta 6i + \beta 7i + \varepsilon i$$

Where $HIV_i$ is the estimated prevalence, (i = 1, 2. . .249), $\beta_0$ is the global intercept, $\beta_s$ are the regression coefficients and the covariates, and $\varepsilon_i$ is the error term:

$\beta 1i$ = weighted proportion of females

$\beta 2i$: weighted average age

$\beta 3i$: weighted proportions of widowed or divorced individuals

$\beta 4i$: weighted proportion of men who are not circumcised

$\beta 5i$: weighted proportion of women-headed households

$\beta 6i$: weighted proportion of food-insecure households

$\beta 7i$: weighted proportion of individuals who had first sexual encounter before age 15 years

OLS results are unreliable when two or more variables exhibit multicollinearity, but GWR builds a local regression equation for each feature in the dataset [33]. The OLS regression model estimates a parameter of interest independent of the location of the particular

observation, whereas the GWR model estimates the parameter of interest under more localized conditions by considering the location of that observation [34]. Therefore, the GWR model is advantageous over OLS when dealing with such spatial datasets, so we used the GWspatial lag regression model to show how the covariate relationships changed in each woreda/town. For all the statistical analysis, statistical significance was decided at p<0.05. Multicollinearity was assessed through variance inflation factor (VIF) and condition number [35], where VIF values greater than 10 indicated multicollinearity [35].

## Ethical considerations

The survey protocol was approved by the Institutional Review Boards of the Ethiopian Public Health Institute (EPHI, Ethiopia), Centers for Disease Control and Prevention (Atlanta, GA USA), and Columbia University (New York, NY USA). The EPHIA Data Analysis Advisory Committee at the EPHI approved the analysis of the data.

## Results

### Characteristics of the study population

Table 1 provides the characteristics of the study population.

Of 11,581 eligible households, 90.9% completed the household interview. We found significantly higher HIV prevalence among women (4.1%) than men (1.9%; p<0.0001), among all participants aged 35–44 years (6.2%) than those aged 15–24 years (0.7%; p<0.0001), and among those with no education (5.2%) than those with more than secondary education (1.0%; p<0.0001). Similarly, women-headed households (4.0%) had higher HIV prevalence than men-headed households (2.2%; p<0.0001). Widowed individuals (14.7%) had significantly higher HIV prevalence than individuals who never married (1.0%; p<0.0001). Participants who had sex before age 15 years (6.8%) had a significantly higher prevalence of HIV than those who had sex at age >15 years (2.9%; p<0.0001 compared with their counterparts (Table 1).

### Geographic distribution of HIV in Ethiopia

HIV prevalence among adults in Ethiopia was 3.0%, which significantly varied by administrative regions (p<0.0001) and ranged from 0.8% in Somali to 5.7% in Gambela (Fig 2).

Of the 249 woredas/towns included in this analysis, 167 (67.1%) had at least one HIV-positive resident. The crude estimates of HIV prevalence among these 167 woredas/towns ranged from 0.6% in Hosana to 25.1% in Meki (Fig 3).

### Spatial heterogeneity of HIV prevalence in urban Ethiopia

HIV prevalence was spatially auto correlated (Moran I, 0.057; p = 0.00014). The interpolated data show the spatial heterogeneity of HIV prevalence (range, 0%–25%), independent of regional boundaries. A higher HIV prevalence was observed in Gambela region, Center-North, and some parts of Eastern Ethiopia. In contrast, HIV prevalence was spatially low in the Center-West and Southern regions (Fig 4). Because EPHIA did not include the eastern parts of the Somali administrative region, the HIV prevalence for this region could not be interpolated in this analysis.

### Spatial determinants of HIV infection in urban Ethiopia

Table 2 describes the spatial determinants of HIV infection in urban Ethiopia.

Compared with the woredas/towns with low HIV prevalence, there was a disproportionately higher rate of men who were not circumcised in Gambela; high HIV burden woredas/

**Table 1. Demographic, socioeconomic, and behavioral characteristics of adults aged 15–64 years by HIV status in urban Ethiopia (N = 19,136), Ethiopia Population-based HIV Impact Assessment Survey (2017–2018).**

| Background characteristics | N (Weighted %) | HIV status | | P-value |
| --- | --- | --- | --- | --- |
| | | HIV negative | HIV positive | |
| | | Weighted % (95% CI) | Weighted % (95% CI) | |
| Women | 11,599 (50.5) | 95.9 (95.5–96.3) | 4.1 (3.7–4.5) | <0.0001* |
| Men | 7,537 (49.5) | 98.1 (97.7–98.4) | 1.9 (1.6–2.3) | |
| **Age, years** | | | | |
| 15–24 | 7,547 (34.9) | 99.3 (99.0–99.5) | 0.7 (0.5–1.0) | <0.0001* |
| 25–34 | 5,664 (30.3) | 97.4 (96.9–97.8) | 2.6 (2.2–3.1) | |
| 35–44 | 3,136 (18.9) | 93.8 (92.8–94.6) | 6.2 (5.4–7.2) | |
| 45–54 | 1,651 (10.1) | 93.9 (92.5–95.1) | 6.1 (4.9–7.5) | |
| 55–64 | 1,138 (5.8) | 96.6 (95.3–97.6) | 3.4 (2.4–4.7) | |
| Man-headed HH | 9,343 (53.7) | 97.8 (97.4–98.1) | 2.2 (1.9–2.6) | <0.0001* |
| Woman-headed HH | 9,793 (46.3) | 96.0 (95.5–96.4) | 3.6 (3.6–4.5) | |
| **Marital status** | | | | |
| Never married | 7,103 (35.5) | 99.0 (98.7–99.3) | 1.0 (0.7–1.3) | <0.0001* |
| Married or living together | 9,418 (52.1) | 97.2 (96.8–97.6) | 2.8 (2.4–3.2) | |
| Divorced or separated | 1,723 (8.5) | 92.3 (90.7–93.5) | 7.7 (6.5–9.3) | |
| Widowed | 7,723 (0.9) | 85.3 (82.3–87.9) | 14.7 (12.1–17.7) | |
| **Education level** | | | | |
| No education | 200 (11.9) | 94.8 (93.7–95.8) | 5.2 (4.2–6.3) | <0.0001* |
| Primary | 6,803 (35.5) | 95.8 (95.2–96.3) | 4.2 (3.7–4.8) | |
| Secondary | 5,488 (28.7) | 97.6 (97.1–98.0) | 2.4 (2.0–2.9) | |
| More than secondary | 4,376 (24.0) | 99.0 (98.6–99.3) | 1.0 (0.7–1.4) | |
| **Food insecurity in the past 4 weeks** | | | | |
| No | 18,223 (95.5) | 97.1 (96.8–97.3) | 2.9 (2.7–3.2) | <0.0001* |
| Yes | 808 (4.5) | 95.0 (93.2–96.3) | 5.0 (3.7–6.8) | |
| **Age at first sex** | | | | |
| First sex at age ≥15 years | 17,735 (95) | 97.1 (96.9–97.4) | 2.9 (2.6–3.1) | <0.0001* |
| First sex at age <15 years | 1,014 (5.0) | 93.2 (91.3–94.7) | 6.8 (5.3–8.7) | |
| Total | 19,136 | 97.0 (96.7–97.2) | 3.0 (2.8–3.3) | |

Abbreviations: $X^2$, chi-square statistics; CI, confidence interval; HH, household.

*P-values of <0.0001 indicate, statistically significant.

towns also had more participants who were not virally suppressed and who were divorced/separated/widowed. In hotspot woredas/towns, the average increase in HIV prevalence among participants who were divorced/separated/widowed was 0.16 and among uncircumcised men was 0.10 (Table 2).

The GWR model is a better fit because it explains 33.4% of total variation in HIV prevalence for the seven variables compared with the OLS model, which explains 25% of the total variation.

## Discussion

Prevalence of HIV infection among adults in Ethiopia was 3.0% (women, 4.1%; men, 1.9%). This corresponds to approximately 384,000 adults living with HIV in urban Ethiopia [4]. Our study estimated the clustering effect of HIV in urban Ethiopia. The crude estimates of HIV prevalence ranged from 0.6% to 25.1%, showing that the HIV prevalence data were spatially

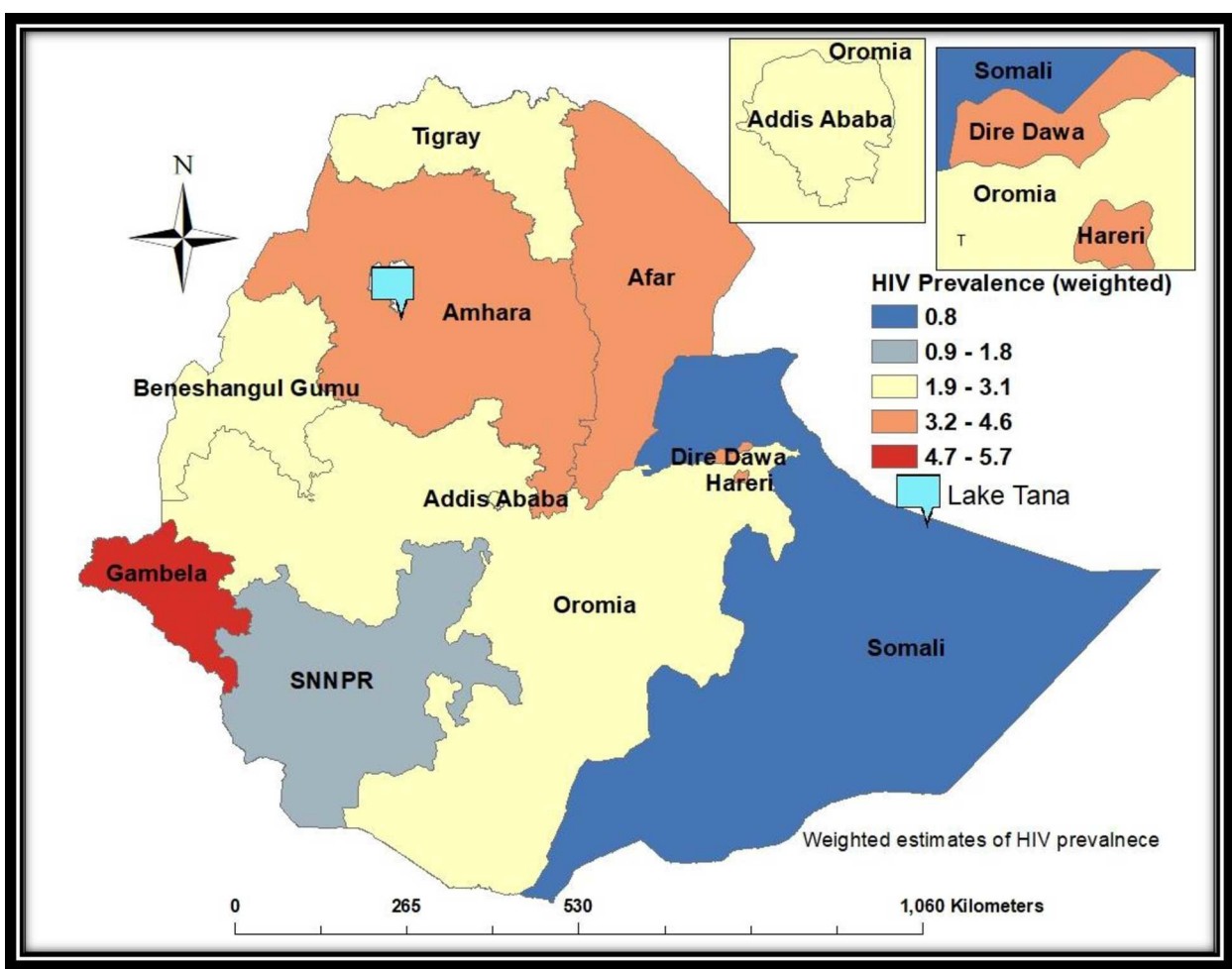

**Fig 2. Weighted distribution of HIV prevalence (%) by region, Ethiopia Population-based HIV Impact Assessment Survey (2017–2018).**
*Area boundaries indicate administrative regions in Ethiopia.

auto correlated (Moran I, 0.057; p = 0.00014). The data indicate that there is <1% likelihood that this cluster pattern of HIV prevalence could be the result of random chance. The spatial analysis helped to identify individuals at high risk of HIV infection due to their geographic location in Ethiopia. The interpolated data showed that there was spatial heterogeneity of HIV prevalence (range, 0%–25%) independent of regional boundaries, which is consistent with previous findings [1, 6, 16, 17, 20, 21, 28, 36, 37]he prevalence of HIV was 10%–21% in certain geographic clusters of Ethiopia [6], which is consistent with findings in other countries [1, 16, 17, 20, 21, 28, 36, 37] indicating a complex geographical variation of the HIV epidemic at the local level.

Our findings show the importance of considering geographic factors in addition to determining individual risk factors to HIV infection in Ethiopia. Previous studies have estimated HIV prevalence in Ethiopia at the national level, disaggregated by urban, rural, or regional levels [10, 38, 39], and most of these studies used blood samples drawn from pregnant women in antenatal facilities [40] and dried blood spot samples from adults [41]. These previous studies provided proximate estimates of prevalence in the overall urban population.

The HIV prevalence maps generated in this analysis describe the spatial disparities in the HIV epidemic within Ethiopia and identify areas with concentrated HIV burden and drivers

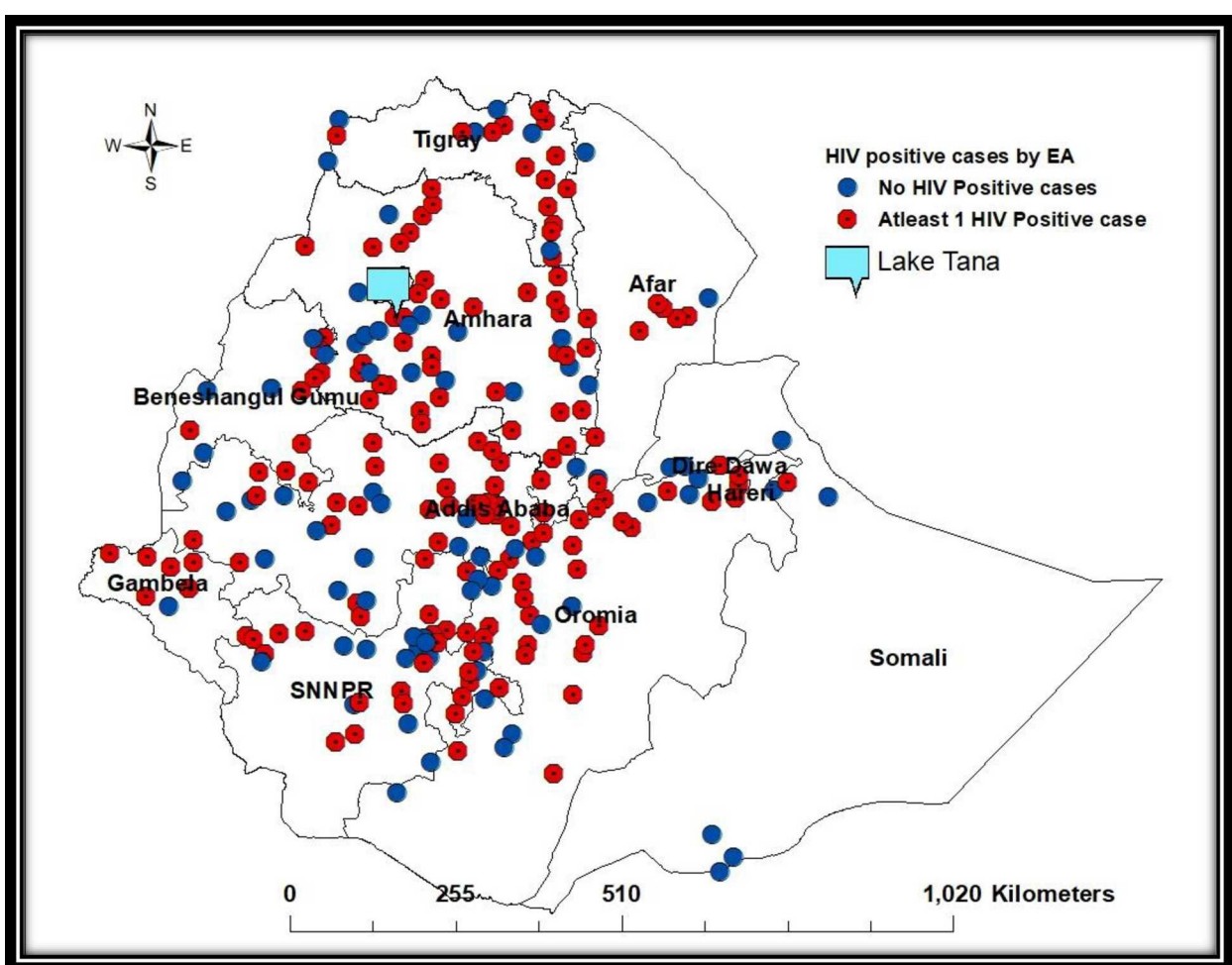

**Fig 3. Distribution of HIV positive cases by enumeration areas, Ethiopia Population-based HIV Impact Assessment Survey (2017–2018).**

of the HIV epidemic. Hotspot analysis results indicate that the current epidemic is concentrated in the Gambela region, in the Center-North region, and in some parts Eastern Ethiopia. The new hotspot in northern Ethiopia calls for further analysis to explore whether HIV prevalence is static or increasing in these areas.

Our findings suggest that localized and woreda/town-specific intervention strategies could help combat the spread of HIV/AIDS in Ethiopia. Prioritizing people at greatest risk of infection and locations with high HIV burden and adapting interventions to reflect the local epidemiological context could increase the efficiency and effectiveness of HIV prevention programs [42]. Our observation that urban individuals who are not virally suppressed, who are divorced/widowed/separated, or who are not circumcised had high HIV infection prevalence in the hotspot woredas/towns concurs with findings of other studies [20, 43–45].

District-based spatial clustering of the HIV epidemic can target interventions at populations with high transmission [46]. Spatial analysis enables policy makers to identify towns most affected and design effective and culturally acceptable preventive measures such as circumcision in the Gambela region, as suggested by other studies [42, 47]. Specific interventions targeted at woredas/towns are more appropriate than universal HIV reduction strategies, which may not be applicable to different cultural contexts in Ethiopia.

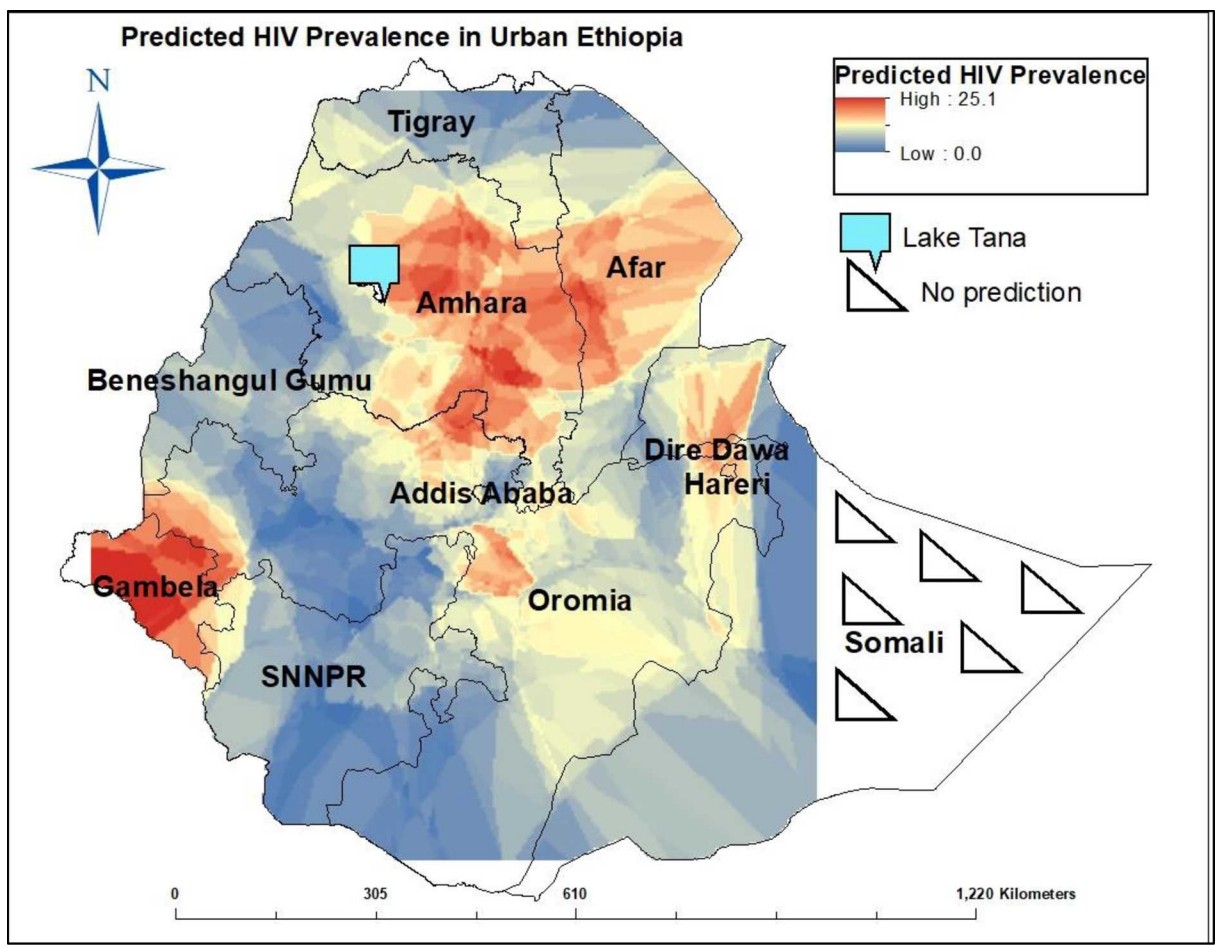

**Fig 4. Predicted HIV prevalence among adults aged 15–64 years in urban Ethiopia, Ethiopia Population-based HIV Impact Assessment Survey (2017–2018).**

Our study also showed high HIV prevalence among women, which is consistent with other studies conducted in Ethiopia [6, 38] and with reports from other countries [1, 19, 23, 28, 45].

**Table 2. Spatial determinants of HIV infection using ordinary least square among adults aged 15–64 years in urban Ethiopia, Ethiopia Population-based HIV Impact Assessment Survey (2017–2018).**

| Variable | Coefficient | Standard Error | Robust P-value | VIF** |
|---|---|---|---|---|
| Intercept | -4.10 | 2.653 | 0.1238 | -------- |
| Women | 0.01 | 0.033 | 0.7873 | 1.73 |
| Age | 0.08 | 0.105 | 0.4438 | 1.29 |
| Divorced/widowed | 0.16 | 0.047 | 0.0007* | 1.48 |
| Not circumcised (only for men) | 0.10 | 0.027 | 0.0005* | 1.11 |
| Women-headed households | 0.02 | 0.016 | 0.1522 | 1.50 |
| Food-insecure household | 0.06 | 0.040 | 0.1590 | 1.07 |
| Age at first sexual encounter (<15 years) | 0.09 | 0.059 | 0.1329 | 1.16 |

Abbreviations: VIF, variance inflation factor.

* P-values <0.05 were considered statistically significant.

** Large VIF values (>7.5) indicate redundancy among explanatory variables.

Our findings indicate that the widowed individual has high HIV prevalence, and 91.9% of this individuals are women. We found that the divorced/widowed individuals in hotspot woredas/ towns had 0.16 average increase in HIV prevalence. The high percentage of widowed women in high HIV prevalence towns might be due to the increased migration of women from rural areas to woredas/towns for job opportunities.

We found that uncircumcised men in hotspot woredas/towns had 0.10 average increase in HIV prevalence. This finding highlights the need to target voluntary medical male circumcision programs in regions such as Gambela, where 61% of the men were circumcised (both medically and traditionally) [48], and less than one-fifth of the circumcisions (18.5%) were performed by health professionals [48].

Targeting regions that have limited capacity to diagnose and treat HIV also could improve outcomes [49]. Our findings contribute to the evidence in the literature and suggest that the national epidemics cannot continue to be assessed as a whole when there is clear evidence of substantial local heterogeneity in the HIV epidemic [50]. The literature from Ethiopia indicates that the overall capacity score (57.1%) for diagnosing and treating HIV in urban facilities was higher than that of the rural health facilities (38.2%) [49]. Our findings, which varied across woredas/towns, also suggest that context specific intervention efforts could more effectively reduce the burden of HIV in Ethiopia.

Our findings are subject to several limitations. We conducted the study in urban areas, so the interpolated data are limited to urban HIV prevalence in Ethiopia. Some regions such as Somali and Benishangul Gumuz had a relatively small number of HIV-positive individuals, which may raise questions related to accuracy of some estimates in these regions. The interpolation of HIV prevalence was not predicted for the six zones in the Somali region. The survey was not powered for analysis at the district level. However, we note that the observed geographical patterns of HIV prevalence parallel that observed in other studies [6], including national surveys such as the demographic and health surveys (DHS) [41] and key population surveillance reports from Ethiopia [40]. This study also has the inherent limitation of a cross-sectional study design, which does not allow for the examination of causal relationships.

The estimates of HIV prevalence across spaces provide an important tool for targeting the interventions that are necessary to bringing HIV infections under control in Ethiopia.

We found higher HIV prevalence in Gambela region, Center-North, and some parts of Eastern Ethiopia. Our findings suggest that uncircumcised men in the certain hotspot towns and divorced/widowed individuals in hotspot woredas/towns might have contributed to the average increase in HIV prevalence in these hotspot areas. Context-specific intervention efforts could decrease the burden of HIV in urban Ethiopia. Localized HIV prevention interventions tailored to at-risk individuals, such as divorced and widowed women and uncircumcised men in certain regions, could be essential to curbing HIV transmission in urban Ethiopia

## Supporting information

**S1 File.**
(DOCX)

## Acknowledgments

We conducted Ethiopia Population-based HIV Impact Assessment Survey through an agreement between Ethiopian Public Health Institute (EPHI), Centers for Disease Control and Prevention (CDC), and ICAP-Columbia University (ICAP-CU). We would like to extend our thanks to the leadership at the Ministry of Health, EPHI, the Regional Health

Bureaus, sub-regional CDC units, and ICAP-CU for their administrative support in organizing and conducting the survey. Our thanks also go to the field coordinators, supervisors, and data collectors for their dedicated work and to all study participants for providing the necessary information.

The EPHIA study group

A list of the study group (survey investigators) is available from: https://phia.icap.columbia.edu/ethiopia-final-report/; and also uploaded as S1 File.

## Author Contributions

**Conceptualization:** Terefe Gelibo, Sileshi Lulseged, Frehywot Eshetu, Saro Abdella, Zenebe Melaku, Solape Ajiboye, Minilik Demissie, Chelsea Solmo, Jelaludin Ahmed, Yimam Getaneh, Ebba Abate.

**Data curation:** Terefe Gelibo, Sileshi Lulseged, Frehywot Eshetu, Zenebe Melaku, Chelsea Solmo, Jelaludin Ahmed, Yimam Getaneh.

**Formal analysis:** Terefe Gelibo, Sileshi Lulseged, Frehywot Eshetu, Chelsea Solmo, Jelaludin Ahmed, Yimam Getaneh.

**Investigation:** Terefe Gelibo, Sileshi Lulseged, Frehywot Eshetu, Saro Abdella, Zenebe Melaku, Solape Ajiboye, Minilik Demissie, Chelsea Solmo, Jelaludin Ahmed, Yimam Getaneh, Ebba Abate.

**Methodology:** Terefe Gelibo, Sileshi Lulseged, Frehywot Eshetu, Saro Abdella, Zenebe Melaku, Solape Ajiboye, Minilik Demissie, Chelsea Solmo, Jelaludin Ahmed, Yimam Getaneh, Susan C. Kaydos-Daniels, Ebba Abate.

**Project administration:** Zenebe Melaku.

**Resources:** Sileshi Lulseged, Zenebe Melaku, Ebba Abate.

**Software:** Terefe Gelibo, Sileshi Lulseged, Frehywot Eshetu.

**Supervision:** Terefe Gelibo, Sileshi Lulseged, Frehywot Eshetu, Saro Abdella, Zenebe Melaku, Solape Ajiboye, Minilik Demissie, Chelsea Solmo, Jelaludin Ahmed, Yimam Getaneh, Ebba Abate.

**Validation:** Terefe Gelibo, Sileshi Lulseged, Frehywot Eshetu, Saro Abdella, Zenebe Melaku, Solape Ajiboye, Minilik Demissie, Chelsea Solmo, Jelaludin Ahmed, Yimam Getaneh, Susan C. Kaydos-Daniels, Ebba Abate.

**Visualization:** Terefe Gelibo, Sileshi Lulseged, Frehywot Eshetu, Saro Abdella, Zenebe Melaku, Solape Ajiboye, Minilik Demissie, Chelsea Solmo, Jelaludin Ahmed, Yimam Getaneh.

**Writing – original draft:** Terefe Gelibo, Sileshi Lulseged, Frehywot Eshetu, Saro Abdella, Zenebe Melaku, Solape Ajiboye, Minilik Demissie, Chelsea Solmo, Jelaludin Ahmed, Yimam Getaneh, Susan C. Kaydos-Daniels, Ebba Abate.

**Writing – review & editing:** Terefe Gelibo, Sileshi Lulseged, Frehywot Eshetu, Saro Abdella, Zenebe Melaku, Solape Ajiboye, Minilik Demissie, Chelsea Solmo, Jelaludin Ahmed, Yimam Getaneh, Susan C. Kaydos-Daniels, Ebba Abate.

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
