## [Decision Letter · Decision Letter 0]

30 Dec 2021

PONE-D-21-34281Spatial distribution and
determinants of HIV prevalence among adults in urban Ethiopia: Findings from the
Ethiopia Population-based HIV Impact Assessment Survey
(2017–2018)PLOS ONE

Dear Dr. Argefa,

Thank you for submitting your manuscript to PLOS ONE. After careful consideration, we
feel that it has merit but does not fully meet PLOS ONE’s publication criteria as it
currently stands. Therefore, we invite you to submit a revised version of the
manuscript that addresses the points raised during the review process.

Please submit your revised manuscript by Feb 13 2022 11:59PM. If you will need more
time than this to complete your revisions, please reply to this message or contact
the journal office at plosone@plos.org. When
you're ready to submit your revision, log on to https://www.editorialmanager.com/pone/ and select the 'Submissions
Needing Revision' folder to locate your manuscript file.

Please include the following items when submitting your revised
manuscript:A rebuttal letter that responds to each point raised by the academic
editor and reviewer(s). You should upload this letter as a separate file
labeled 'Response to Reviewers'.A marked-up copy of your manuscript that highlights changes made to the
original version. You should upload this as a separate file labeled
'Revised Manuscript with Track Changes'.An unmarked version of your revised paper without tracked changes. You
should upload this as a separate file labeled 'Manuscript'.

If you would like to make changes to your financial disclosure, please include your
updated statement in your cover letter. Guidelines for resubmitting your figure
files are available below the reviewer comments at the end of this letter.

We look forward to receiving your revised manuscript.

Kind regards,

Mona Pathak, PhD

Academic Editor

PLOS ONE

Journal Requirements:

“This project has been supported by the U.S. President’s Emergency Plan for AIDS
Relief (PEPFAR) through the U.S. Centers for Disease Control and Prevention (CDC)
under the terms of cooperative agreement #U2GGH001226. The findings and conclusions
in this report are those of the authors and do not necessarily represent the
official position of the funding agencies. Conflict of interest.”

4. We note that Figure 1, 2, 3 and 4 in your submission contain map images which may
be copyrighted. All PLOS content is published under the Creative Commons Attribution
License (CC BY 4.0), which means that the manuscript, images, and Supporting
Information files will be freely available online, and any third party is permitted
to access, download, copy, distribute, and use these materials in any way, even
commercially, with proper attribution. For these reasons, we cannot publish
previously copyrighted maps or satellite images created using proprietary data, such
as Google software (Google Maps, Street View, and Earth). For more information, see
our copyright guidelines: http://journals.plos.org/plosone/s/licenses-and-copyright.

 a. You may seek permission from the original copyright holder of Figure 1, 2, 3 and
4 to publish the content specifically under the CC BY 4.0 license. 

5. Please ensure that you refer to Figure 1 and 2 in your text as, if accepted,
production will need this reference to link the reader to the figure.

Reviewers' comments:

Reviewer's Responses to Questions

**Comments to the Author**

1. Is the manuscript technically sound, and do the data support the conclusions?

Reviewer #1: Yes

Reviewer #2: No

2. Has the statistical analysis been performed
appropriately and rigorously? 

Reviewer #1: Yes

Reviewer #2: No

3. Have the authors made all data underlying the
findings in their manuscript fully available?

Reviewer #1: Yes

Reviewer #2: No

4. Is the manuscript presented in an intelligible
fashion and written in standard English?

Reviewer #1: Yes

Reviewer #2: Yes

5. Review Comments to the Author

Reviewer #1: 

The scope of the study is well designed to delineate hotspot HIV
transmission risk.

The study result cloud be used to control and management of the HIV
infection in the vulnerable community through proper intervention measures target
the gender and affected age groups with efforts of public health officials and
national health programmers

Random Sampling methods for field data collection of HIV epidemiological
information in the selected house hold, and geo-coordinates of the house hold
location are depicted on map, using GPS.

Geo-Spatial analysis was applied significantly

Moran I geo-statistical analysis measure of spatial autocorrelation in
area data, explaining the degree of dependence between values of a variable at
different geographic locations.

Geographically weighted regression (GWR) analysis was used to establish
the relationship of covariates.

The interpolation of spatial prediction of HIV infection using Kriging
methods, and results reflect the real situation of the ground.

Reviewer #2: Please find my comments and suggestions below. I have mainly focused on
the spatial analysis aspects of this study and please read my suggestions below.

Study variables and measurement (Lines 87-98)

Please clearly define your outcome variable. Please define the geo-linked HIV
prevalence, geo-linked to what geographical level? It is not clearly described how
the enumeration areas (EAs) are related to “towns” Does each enumeration area
consist of one town? Or do you need to aggregate several EAs to the town level?

When you define the prevalence of HIV, it is not clear what a cluster means. Does
“cluster” refer to an EA or a town? When you are mentioning that you have calculated
the weighted HIV prevalence in each town, but you did not describe the weighting
method.

In Figure 2 you are presenting a prevalence map, but the area boundaries are not
defined. Are those provinces in Ethiopia? Additionally, as you have mentioned just a
small portion of EAs could be sampled from “Somali” district, and for that reason,
you should change the map to include a small portion of that district. Figure 2
implies that HIV prevalence is low in Somali district, but we do not know because
the population was not included in the EPHIA survey.

Line 91 – Please state that the outcome variable for your regression model (OLS) was
the weighted HIV prevalence in each town.

Line 92 – “Study variables” is a confusing term, because study variables could
include the outcome (e.g., dependent) variable and the predictor (e.g., independent)
variables. You should use “predictor” or “independent” variables instead of study
variables.

Analysis (Lines 139-190)

The hotspot analysis (Getis-Ord Gi*statistic) is not a spatial interpolation method.
Moreover, you did not present any local clustering (hot spot) analysis. Please
describe in detail your Kriging method, because there is more than one Kriging
method that you can use. Kriging is a spatial interpolation method and you can not
identify statistically significant clusters of high or low HIV prevalence based on
Kriging analysis. You might confuse Kriging with the Getis-Ord Gi* statistic or the
Local Moran’s I method where you can identify high or low (if you use Getis-Ord Gi*)
and high-high, low-low, low-high, and high-low (if you use Local Moran’s I) HIV
prevalence clusters, but you did not present any of those results.

You can read my previous papers where I have used and described various spatial
epidemiology methods:

https://journals.plos.org/plosone/article?id=10.1371/journal.pone.0235291

https://bmcinfectdis.biomedcentral.com/articles/10.1186/s12879-015-1106-6

Line 148 - Global Moran’s I statistic evaluates the extent of spatial heterogeneity.
You should define at what distance band did you observe the highest Moran's I value
(see my papers for the methods). Please also define what type of conceptualization
parameter did you use, because you can use more than one (e.g., zone of
indifference, inverse distance, etc.). I think that you should use and present local
Moran’s I analysis to identify local HIV clusters.

Line 168 – Please define your GWR model. What type of spatial weights matrix did you
use? What type of GRW did you use, spatial lag model or spatial error model?

You can read about them at:

https://s4.ad.brown.edu/resources/tutorial/modul2/geoda3final.pdf

Thank you!

6. PLOS authors have the option to publish the peer
review history of their article (what does this mean?). If published, this will
include your full peer review and any attached files.

If you choose “no”, your identity will remain anonymous but your review may still be
made public.

**Do you want your identity to be public for this peer review?** For
information about this choice, including consent withdrawal, please see our
Privacy Policy.

Reviewer #1: **Yes: **Masimalai Palaniyandi., M.Sc.,M.Tech.,Ph.D.,

Reviewer #2: **Yes: **Csaba Varga

Recommendation and Comments for Manuscript Number
PONE.pdf
---

## [Author Response · Author response to Decision Letter 0]

14 May 2022

The responses are incorporated in the manuscript. Double checked the link to data
source and updated it as appropriate.

rebuttal letter .docx
---

## [Decision Letter · Decision Letter 1]

27 Jun 2022

Spatial distribution and determinants of HIV prevalence among adults in urban
Ethiopia: Findings from the Ethiopia Population-based HIV Impact Assessment Survey
(2017–2018)

PONE-D-21-34281R1

Dear Dr. Argefa,

We’re pleased to inform you that your manuscript has been judged scientifically
suitable for publication and will be formally accepted for publication once it meets
all outstanding technical requirements.

Kind regards,

Mona Pathak, PhD

Academic Editor

PLOS ONE

Reviewers' comments:

Reviewer's Responses to Questions

**Comments to the Author**

1. If the authors have adequately addressed your comments raised in a previous round
of review and you feel that this manuscript is now acceptable for publication, you
may indicate that here to bypass the “Comments to the Author” section, enter your
conflict of interest statement in the “Confidential to Editor” section, and submit
your "Accept" recommendation.

Reviewer #1: All comments have been addressed

Reviewer #2: All comments have been addressed

2. Is the manuscript technically sound, and do the data
support the conclusions?

Reviewer #1: Yes

Reviewer #2: Yes

3. Has the statistical analysis been performed
appropriately and rigorously? 

Reviewer #1: Yes

Reviewer #2: Yes

4. Have the authors made all data underlying the
findings in their manuscript fully available?

Reviewer #1: Yes

Reviewer #2: Yes

5. Is the manuscript presented in an intelligible
fashion and written in standard English?

Reviewer #1: Yes

Reviewer #2: Yes

6. Review Comments to the Author

Reviewer #1: (No Response)

Reviewer #2: (No Response)

7. PLOS authors have the option to publish the peer
review history of their article (what does this mean?). If published, this will
include your full peer review and any attached files.

If you choose “no”, your identity will remain anonymous but your review may still be
made public.

**Do you want your identity to be public for this peer review?** For
information about this choice, including consent withdrawal, please see our
Privacy Policy.

Reviewer #1: **Yes: **Dr.Palaniyandi Masimalai

Reviewer #2: **Yes: **Csaba Varga

---

## [Editor Report · Acceptance letter]

1 Jul 2022

PONE-D-21-34281R1 

Spatial distribution and determinants of HIV prevalence among adults in urban
Ethiopia: Findings from the Ethiopia Population-based HIV Impact Assessment Survey
(2017–2018) 

Dear Dr. Gelibo:

I'm pleased to inform you that your manuscript has been deemed suitable for
publication in PLOS ONE. Congratulations! Your manuscript is now with our production
department. 

Kind regards, 

on behalf of

Dr. Mona Pathak 

Academic Editor

PLOS ONE